# *FADD* in Cancer: Mechanisms of Altered Expression and Function, and Clinical Implications

**DOI:** 10.3390/cancers11101462

**Published:** 2019-09-29

**Authors:** José L Marín-Rubio, Laura Vela-Martín, José Fernández-Piqueras, María Villa-Morales

**Affiliations:** 1Institute for Cell and Molecular Biosciences, Newcastle University, Newcastle upon Tyne NE2 4HH, UK; Jose.Marin-Rubio@newcastle.ac.uk; 2Departamento de Biología, Universidad Autónoma de Madrid, 28049 Madrid, Spain; laura.velam@estudiante.uam.es; 3Centro de Biología Molecular Severo Ochoa (CBMSO), 28049 Madrid, Spain; 4Centro de Investigaciones Biomédicas en Red de Enfermedades Raras (CIBERER), 28029 Madrid, Spain; 5IIS-Fundación Jiménez Díaz, 28040 Madrid, Spain

**Keywords:** *FADD*, chromosomal alterations, mutations, polymorphisms, transcription factors, epigenetic regulation, posttranslational modifications, gene expression, metabolism, clinical implications

## Abstract

FADD was initially described as an adaptor molecule for death receptor-mediated apoptosis, but subsequently it has been implicated in nonapoptotic cellular processes such as proliferation and cell cycle control. During the last decade, FADD has been shown to play a pivotal role in most of the signalosome complexes, such as the necroptosome and the inflammasome. Interestingly, various mechanisms involved in regulating FADD functions have been identified, essentially posttranslational modifications and secretion. All these aspects have been thoroughly addressed in previous reviews. However, FADD implication in cancer is complex, due to pleiotropic effects. It has been reported either as anti- or protumorigenic, depending on the cell type. Regulation of *FADD* expression in cancer is a complex issue since both overexpression and downregulation have been reported, but the mechanisms underlying such alterations have not been fully unveiled. Posttranslational modifications also constitute a relevant mechanism controlling FADD levels and functions in tumor cells. In this review, we aim to provide detailed, updated information on alterations leading to changes in *FADD* expression and function in cancer. The participation of FADD in various biological processes is recapitulated, with a mention of interesting novel functions recently proposed for FADD, such as regulation of gene expression and control of metabolic pathways. Finally, we gather all the available evidence regarding the clinical implications of *FADD* alterations in cancer, especially as it has been proposed as a potential biomarker with prognostic value.

## 1. Regulation of *FADD* Gene Expression

### 1.1. Genetic and Chromosomal Alterations Affecting FADD

The *FADD* gene is located on chromosome 11q13.3 in humans, and consists of two exons separated by a 2-kb intron and only one isoform. *FADD* is expressed in every adult and embryonic tissue in mice and humans [1,2] (Figure 1).

*FADD* expression is altered in many cancer types. This is, however, a controversial issue, since both overexpression [4,5,6,7,8,9,10] and downregulation [11,12,13,14,15] have been observed, depending on the cancer type. According to The Human Protein Atlas, FADD protein levels are low in most normal tissues. In cancer, *FADD* is detected in all tumor types analyzed by RNA sequencing according to The Cancer Genome Atlas (TCGA), and FADD protein levels obtained by immunohistochemistry with two different antibodies reveal results that are consistent with RNA-seq and/or protein/gene characterization data (Figure 2). A variety of FADD protein levels can be observed across the different tumor types; the mechanisms responsible for altered expression are, however, not always elucidated.

A major event leading to overexpression is DNA amplification. Chromosome 11q13.3 is a 1.7 Mb-region that contains 12 other genes apart from *FADD* [7]. Amplification of this region has been frequently found in human cancers and is associated with poor prognosis. Among the candidate genes within the region, *FADD* has attracted interest since it was described as the only gene in the minimum region of overlap within the amplification in a series of squamous cell carcinomas of the head and neck (HNSCC) [16]. Furthermore, this group and others have demonstrated the association between *FADD* amplification and high FADD levels with poor overall survival and disease-free survival, especially if both events occur together [6,17,18]. This supports the notion that FADD is a driver of the tumorigenic effects of the 11q13.3 amplification. Based on the results from the recent TCGA PanCancer Atlas Studies (https://www.cbioportal.org/results/cancerTypesSummary?case_set_id=all&gene_list=FADD&cancer_study_list=5c8a7d55e4b046111fee2296), with 10,953 patients and 10,967 tumor samples from various cancer types, DNA amplification is by far the most frequent event affecting FADD and is most prominent in esophageal squamous cell carcinoma and HNSCC, with nearly 60% and 30% of cases harboring the amplification, respectively [19,20,21,22,23,24]. Other cancer types showing FADD-containing chromosomal amplification of 11q13 region are breast cancer [25,26,27], bladder cancer [28,29,30,31], lung squamous cell carcinoma [32], and, less frequently, ovarian cancer, where the region commonly involved is more distal [33,34,35].

Nevertheless, increased FADD levels do not always coexist with *FADD* gene amplification, as reported in acute myelogenous leukemia [36] or lung adenocarcinoma [5,37], suggesting the occurrence of additional mechanisms capable of modulating FADD levels in cancer.

Allelic losses or chromosomal deletions might account for *FADD* downregulation. However, the results compiled in databases (cBioPortal) point to this being a very infrequent event affecting *FADD*. For a long time, allelic losses within 11q13 have been observed in several cancer types, but the analyses have focused on a more proximal region of the chromosome [38,39,40,41,42]. An association between isolated familial somatotropinoma (IFS) and loss of heterozygosity (LOH) on a 2.21 Mb-region of chromosome 11q13 has been well established, although the driver gene has not been identified [43]. Deletions and LOH encompassing *FADD* gene within 11q13 have been found in 25% (9/36) of cervical cancer samples [44], although *FADD* was not identified as a driver of the potential LOH-derived tumorigenic effects. In a cohort of 60 cases of non-small-cell lung cancer (NSCLC), LOH at chromosome 11q13.3 was found in around 20% of the samples [45]. Hemizygosity affecting *FADD* has also been reported in ocular coloboma [46], a manifestation of otodental syndrome or, as more recently re-named, the chromosome 11q13 deletion syndrome [47].

Data from TCGA PanCancer Atlas Studies show a correlation between *FADD* expression, as determined by RNA sequencing, and copy number alterations (Figure 3).

Apart from cancer, allelic losses or gains involving *FADD* have been associated with developmental disabilities or congenital anomalies (specifically, global developmental delay for ClinVar Variation ID 153941; global developmental delay and abnormal heart morphology for ClinVar Variation ID 442080; abnormality of the ear, obsolete malformation of the heart and great vessels, short stature and polydactyly for ClinVar Variation ID 441904; and micrognathia, syndactyly, intrauterine growth retardation, ventricular septal defect and abnormal facial shape for ClinVar Variation ID 441903) [48].

Mutations in *FADD* might also account for altered expression. However, *FADD* mutations are not frequent in cancer. According to the COSMIC v89 (Catalogue of Somatic Mutations in Cancer) database, only 48 gene variants have been found in 56 out of 36187 (0.15%) tested tumor samples to date. Of them, 11 are synonymous and 37 affect the protein sequence either by nucleotide substitution or deletion. This concurs with the results gathered by the aforementioned TCGA PanCancer Atlas Studies, which showed 0.4% of somatic mutations in *FADD* and a total number of 47 different mutations, 41 of them affecting the protein sequence and six involving FADD in fusion proteins (FADD mutations, TCGA PanCancer Atlas, cBioPortal) (Figure 4A). The IntOGen-mutations platform (http://www.intogen.org/mutations/) identifies cancer drivers according to specific criteria [49]. To date, it has only compiled 12 *FADD* mutations, two of them synonymous (Figure 4B), and none of them is catalogued as a driver.

Particular studies confirm that *FADD* mutation is indeed a rare event in cancer. In NSCLC, four missense mutations of *FADD* were detected in 80 samples (5%) [50], whereas only one *FADD* mutation was found in 98 colorectal adenocarcinomas (1%) and no *FADD* mutation was detected in 116 advanced gastric adenocarcinomas [51]. Moreover, no mutations in *FADD* were found either in 92 samples from diverse hematological malignancies [52], in 24 lung adenocarcinomas with high *FADD* expression [5], in 15 osteosarcoma tumor samples [53], or in murine (14 samples) [12] and human (22 samples) [13] T-cell lymphoblastic lymphoma (T-LBL).

Contrary to the correlation observed between *FADD* expression and copy number alterations (Figure 3), no specific relationship between *FADD* expression and mutations can be deduced according to the data from TCGA PanCancer Atlas Studies (Figure 5).

Polymorphisms in *FADD* are numerous. To date, 816 variants have been described (Ensembl Release 97). However, only seven of them have had clinical consequences attributed, and these are not specifically related to any tumor type (Table 1) [54,55].

Among the 816 variants, the specific consequence of 135 of these variants is to affect the regulatory region of *FADD*, although none of them is catalogued in ClinVar due to reported clinical significance. Global minor allele frequency (MAF), defined in humans by the 1000 Genomes Project phase 3, is only available for 17 of them. According to this information, only two variants can be considered as common (MAF ≥ 5%), specifically rs12295430 (global MAF = 0.053) and rs41268205 (global MAF = 0.051). Interestingly, no publication has reported any functional impact associated with their occurrence. The remaining 15 are rare variants, with very low global MAF values (<0.001 for 13 of them). Among the 816 variants, 63 are reported as affecting transcription factor binding sites; only one of them (rs10898853) exhibits a minor allele frequency above rare variance (5%), specifically of 0.318. In particular, this variant has been associated with susceptibility to papillary thyroid cancer [56].

Polymorphisms reported in murine *Fadd* [57] did not show evident functionality. Regarding humans, the recent GTEx Project (GTEx Analysis Release V8, dbGaP Accession phs000424.v8.p2) identifies variations in gene expression that are highly correlated with genetic variation (the so-called eQTL, expression quantitative trait loci). To date, eighty-two functional polymorphisms predicted to influence tissue-specific *FADD* expression have been identified. However, according to the interpretation of the data, a limited number of variants are predicted to have an eQTL effect, specifically those with *m*-values > 0.9 (e.g., the three dots on the right in Figure 6).

In summary, *FADD* expression is frequently altered in cancer and its increase is most often related to DNA amplification. However, although allelic losses affecting *FADD* usually lead to decreased expression, the latter is not always explained by copy number alteration, nor is it by mutations or functional polymorphisms since these are infrequent.

### 1.2. Transcription Factors Affecting FADD

The 5’ terminus of *FADD* gene was first studied by Kim and colleagues [59], who reported a 1-kb region containing consensus sequences for regulatory elements such as Lyf-1, a reverse copy of the core element of the insulin enhancer (EI), AP-1, N-Myc and SP-1; a TATA-box was identified 620 bp upstream of the translation start site. Much more recently, the signals of DNase hypersensitivity together with histone modification H3K4me3 have been used to predict promoter-like regions by ENCODE 4 consortium. This method has been applied to 107 human cell types and 14 mouse cell types with both DNase and H3K4me3 data generated by the ENCODE and Roadmap Epigenomic consortia. Transcription factor binding sites within promoter-like regions have been proposed based on of ChIP-seq data, generating peaks for 161 transcription factors in 91 cell types [60]. Regarding *FADD*, these transcription factor binding sites have been identified in the promoter region (UCSC Genome Browser); however, very few publications have so far reported the role of specific transcription factors on regulating *FADD* expression. One of them demonstrates that HIF-1α binds to and represses the *FADD* promoter [61]. Since HIF-1α is overexpressed in many cancers and has been associated with tumor aggressiveness and poor prognosis [62], it could be speculated that *FADD* downregulation would be involved. Another study [63] reports the interaction with and activation of *FADD* promoter by BRCA1, suggesting that *BRCA1* loss or inactivation in a tumor can cause reduced levels of *FADD* that in turn desensitize cells to apoptosis. Hence, the authors propose *FADD* as a candidate biomarker for *BRCA1*-associated breast cancer. In addition, the PAX2 and VAX2 transcription factors have been reported to co-regulate *fadd* transcriptional activation in zebrafish [64]. As mentioned before, LOH affecting *FADD* was reported in ocular coloboma [46]; since *PAX2* mutations have been identified as one of the most common causes of coloboma [65], it could be speculated that reduced *FADD* expression—either due to LOH or to mutations in *PAX2* causing regulation of transcription—would play a role in coloboma.

### 1.3. Epigenetic Regulation of FADD Expression

Epigenetic regulation of gene expression is without a doubt a relevant mechanism. Changes in promoter methylation, histone modification, or the action of microRNAs (miRNAs) have been reported with variable frequency as regulatory processes affecting *FADD* levels.

#### 1.3.1. Methylation of *FADD* Promoter Region

DNA methylation in cytosines within CpG islands of gene promoters is a prominent mechanism of gene expression regulation, since hypermethylation is related to inactivation of transcription and the opposite is true for hypomethylation [66,67]. Evidence for such mechanism affecting *FADD* expression is very limited. To date, *FADD* reduction due to hypermethylation of its promoter has been reported in myelodysplastic syndrome [68] and inflammatory processes like apical periodontitis [69]. Regarding cancer, one study reported an association between significant hypermethylation of *FADD* promoter in oral squamous cell carcinoma and reduced *FADD* expression [70,71]. Therefore, further studies on whether *FADD* promoter hypermethylation is a frequent event leading to *FADD* reduction, particularly in cancer, where resistance to apoptosis is a hallmark, would be of interest. If confirmed, *FADD* promoter hypermethylation could be tested as a prognostic factor.

#### 1.3.2. Histone Modification of *FADD*

Histones undergo reversible, dynamic and multiple modifications at their basic amino acids, such as methylation, acetylation and phosphorylation. These modifications constitute the histone code, which can be implicated in the regulation of gene expression among other functions. As they can change the structure of chromatin, some epigenetic modifications are associated with active transcription, such as acetylation, while others are related with repression of transcription [67,72]. During recent years, interesting bioinformatics tools for prediction of histone modification and related function have been developed [73]. An in silico search for predicted histones marks affecting *FADD* using CHIP-seq data deposited in ENCODE (SCREEN hg19) revealed 12 candidate cis-regulatory elements (ccREs) located between the first and last transcription start sites of *FADD* plus 50kb upstream (Figure 7) [74,75]. In addition to the four core epigenomic marks used to generate the registry (representative DNase hypersensitivity sites, H3K4me3, H3K27ac, and CTCF ChIP-seq signals) all available histone marks and transcription factor ChIP-seq peaks are annotated, identified using ENCODE uniform processing pipelines. Each of the 12 candidate elements for *FADD* shows significant signals for many different histone marks (up to 31 in the candidate with the highest scores), but these data are not supported by any experimental evidence in the literature. Moreover, scarce evidence of histone modifications directly affecting *FADD* expression is available. It has been proposed that the inhibitor of growth 1 (ING1) may contribute to regulate acetylation levels of different histones, thus promoting transcriptional activation of components of the apoptotic pathway, such as *FADD* [76]. The authors propose that *ING1* downregulation in glioblastoma would result in *FADD* suppression, leading to apoptosis resistance in cancer cells upon treatment with histone deacetylase inhibitors (HDACi). The expression of *Fadd* was found to be suppressed in murine leukemia expressing the aberrant RUNX1-EVI1 generated by t(3;21) [77]. RUNX1-EVI1 recruits histone deacetylase, thus resulting in transcriptional dysregulation of wild-type RUNX1-target genes. *Fadd* bears binding sites for RUNX1, so RUNX1-EVI1-driven *Fadd* deacetylation is a plausible explanation for *Fadd* repression in these tumors. Interestingly, the authors demonstrate the restoration of *Fadd* expression upon treatment with HDACi, proposing that such inhibitors could be useful for leukemia patients expressing RUNX1-EVI1.

#### 1.3.3. microRNAs Controlling *FADD*

Gene silencing by microRNAs is another prominent mechanism of regulation, combining translational repression and mRNA destabilization of target genes [78]. According to miRTarBase, there are 20 miRNAs that putatively target murine *Fadd*. Of them, only mmu-miR-134-5p and mmu-miR-155-5p have been reported with strong evidence [79,80,81]. In humans, miRTarBase only contains seven putative miRNAs targeting *FADD*. Again, only two of them have been proven with strong evidence, hsa-miR-155-5p and hsa-miR-128-3p [82,83,84]. Thus, miR-155-5p is the only microRNA strongly demonstrated to regulate *FADD* both in mice and humans. In mice, *mmu-miR-155-5p* is upregulated in response to lipopolysaccharide and subsequent silencing of *Fadd* promotes an anti-apoptotic effect [80]. A similar effect has been observed in RAW264.7 cells, where *mmu-miR-155-5p* upregulation leads to *Fadd* silencing and, in consequence, to reduced apoptosis [81]. In the human gastric epithelium and mucosa, *FADD* silencing due to *hsa-miR-155-5p* overexpression induced upon *Helicobacter pylori* infection promotes the activation of an inflammatory response [83]. In contrast, *hsa-miR-155-5p* is downregulated in degenerative nucleus pulposus, leading to *FADD* increase, which promotes high levels of apoptosis and eventually intervertebral disc degeneration [82]. Its alterations in cancer are reviewed in [85]: *miR-155* is upregulated in most hematological malignancies, such as B cell lymphomas, acute myelomonocytic leukemia, and acute monocytic leukemia, as well as in solid tumors, such as breast, colon, and lung cancers. However, a direct link between *miR-155* overexpression and *FADD* downregulation in those tumor samples has not been demonstrated. Interestingly, the level of *miR-155* expression can be used to distinguish between germinal center B cell-like and activated B cell-like subtypes of diffuse large B cell lymphoma, so its convenience for diagnosis has been suggested.

## 2. Posttranslational Modifications of FADD Protein

The discrepancy between genetic alterations affecting *FADD* and protein levels suggests that the latter would be regulated by mechanisms affecting the stability of the protein, such as posttranslational modifications (PTMs) [6]. For example, it was demonstrated that transcription was not altered in samples from acute myelogenous leukemia exhibiting reduced FADD protein [36]. Previously, the same authors had proposed in the adenomatous or adenocarcinomatous mouse thyroid gland a non-proteasome mediated loss of FADD protein [15].

Different PTMs have been reported to affect the functions of FADD, but the most important and well-studied is phosphorylation [86,87,88]. Several phosphorylation sites have been identified in FADD [89,90,91], but the most relevant site conserved between mice and humans is serine 191 and serine 194, respectively [90]. The impact of FADD phosphorylation on apoptosis is controversial [5,90,92,93]. Our data indicate that the phosphorylation status of FADD does not affect apoptosis [13], but FADD phosphorylation could affect apoptosis indirectly, as a consequence of protein availability in the cell. It has been suggested that phosphorylated FADD normally resides in the nucleus and that activation with Fas or TNFα/ActD reduces its phosphorylation and redistributes unphosphorylated FADD to the cytoplasm, contributing to the efficient propagation of cell death signaling [94,95,96,97]. Phosphorylation in this serine has been reported to play a very important role in regulating cell growth and proliferation [37,88,90,98,99]. Moreover, the deregulation of FADD phosphorylation at serine 194 has been identified as a relevant clinical issue in several types of hematological [13,100,101] and solid cancers [5,7,37,88,90,98,99,102,103]. A molecular mechanism associating the increase in phosphorylated FADD with tumorigenesis has been hinted at only in limited studies. In lung cancer, it has been demonstrated that increased levels of nuclear phosphorylated FADD induce NF-κB, which is suggested to induce the expression of cyclin D1 (*CCND1*) at the transcriptional level, ultimately resulting in cell cycle deregulation [5]. Also in lung cancer, the induction of mitosis seems to depend on activation of KRAS and CK1α phosphorylation of FADD during G2–M, where FADD interacted with G2–M cell-cycle regulatory components PLK1, AURKA, and BUB1 [102]. In breast cancer, AK2 downregulation results in reduced interaction of AK2-DUSP26 complex with FADD and, subsequently, increased phosphorylated FADD levels in the nuclei of tumor cells, correlating with increased proliferation [98].

There is limited information regarding FADD regulation by PTMs other than phosphorylation, but the available evidence indicates that ubiquitination regulates FADD stability and, consequently, the ability of FADD to mediate apoptosis [104]. Interestingly, both modifications could be related. Very recently, we proposed that FADD phosphorylation stabilizes the protein and that its degradation is mediated by the proteasome [13]. However, a direct link between FADD phosphorylation status, FADD ubiquitination, and proteasome-mediated degradation has not been established so far. It has been described that E3-ubiquitin ligase Makorin RING Finger Protein 1 (MKRN1) induces K48-linked polyubiquitination of FADD and, therefore, its degradation by the proteasome [105]. However, it cannot be ruled out that other E3 ligases could also ubiquitinate FADD [104]. Eight lysines are susceptible to ubiquitination in FADD (K24, K33, K35, K110, K120, K125, K149, and K153). K6-linked polyubiquitination at residues K149 and K153 by C terminus HSC70-interacting protein (CHIP) has been reported to prevent the formation of the death-inducing signaling complex (DISC) and therefore to suppress the apoptosis mediated by Fas or DR4/DR5, although it does not affect FADD degradation [106]. On the other hand, the linear ubiquitin chain assembly complex (LUBAC), which is composed of HOIP, HOIL-1L, and SHARPIN subunits, specifically generates M1-linked linear polyubiquitination of FADD and NEMO upon NF-κB activation, also suppressing cell death [107]. Recently, it has been demonstrated that FADD is ubiquitinated by speckle-type POZ protein (SPOP), which causes proteasome-mediated degradation [108] and impairs the FADD-mediated activation of NF-κB signaling [108,109]. SPOP has been reported to suppress tumorigenesis by inhibiting the NF-κB pathway in certain tumor types [109,110], indicating that this SPOP‒FADD‒NF-κB axis might represent a molecular mechanism underlying the role of FADD alteration in non-small-cell lung cancer [109]. In summary, E3-ubiquitin ligases constitute a potential therapeutic target with regards to FADD availability and function in the tumor cell. The combined treatment with E3-ubiquitin ligase inhibitors and agents able to activate extrinsic apoptosis such as TRAIL or TNFα might be beneficial for certain tumor types.

## 3. The Roles of FADD and Their Participation in Cancer

As a bimodular protein, FADD is a well-known adapter for receptor-induced cell death. Through homotypic interaction of its death domain (DD), FADD can be recruited to death receptors (DRs) belonging to the family of tumor necrosis factor (TNF) receptors including TNF-R1, FAS (CD95/APO-1), DR3, TRAIL (TNF-related apoptosis-inducing ligand)-R1 (DR4), and TRAIL-R2 (DR5), transmitting apoptosis initiating signals by their specific death ligands [1,111,112,113]. The death-effector domain (DED) of FADD recruits DED-only proteins (procaspase-8, procaspase-10, or c-FLIP) to form an active DISC. Mediated both by proximity-induced dimerization and proteolytic cleavage [114], initiator procaspases activate in the DISC and, once activated, caspase-8/caspase-10 initiate the caspase cascade, directly through cleavage of procaspase-3 or indirectly via Bid cleavage, eventually inducing apoptosis. Alternatively, under certain stimuli resulting in the blockade of apoptosis, such as the triggering of viral defense mechanisms or under caspase inhibitory conditions, death receptors signaling may lead to necroptosis, which occurs through a RIPK1‒RIPK3‒MLKL axis [115]. FADD acts as a negative regulator of this process, as it adapts the recruitment of caspase-8/cFLIP to RIPK1/3, which leads to cleavage of their kinase domain and subsequent inactivation [116,117]. The so-called ripoptosome comprises RIPK1/FADD/caspase-8 and induces apoptosis in a ligand/receptor-independent manner as it can be induced upon genotoxic stresses such as etoposide or removal/inhibition of the inhibitor of apoptosis (IAP) protein family; recruitment of RIPK3 to the complex switches the ripoptosome towards necroptosis [105,118]. In cancer, apoptosis impairment appears as a plausible consequence of FADD reduction [87,119]. Moreover, as FADD can prevent necroptosis, FADD reduction in cancer would also induce a switch from apoptotic signaling to necroptosis [117].

Additionally, FADD has been proposed to participate in cell cycle progression, T-cell proliferation, survival, or genome surveillance in the nucleus [87,96,120,121]. Several studies have evidenced defects in proliferation and cell cycle progression in FADD-deficient cells [87,101,122,123,124]. Although these effects could be simply attributed to the enhancement of RIPK1-dependent cell death in the absence of FADD, many studies using phosphorylation mutants highlight the relevance of FADD phosphorylation in G2/M phase and its interaction with proteins that are important for cell cycle progression [5,90,92,102,120,124,125,126,127]. Whether FADD participation in these functions is an indirect consequence of being sheltered from death receptors in the nucleus or an actively regulated role remains unclear. FADD can recruit proteins that regulate the NF-κB and MAPK pathways, which promote proliferation and cell cycle progression [5]. Our group has reported a reduction of FADD together with reduced apoptosis in T-LBL samples, but the accumulation of phosphorylated FADD in the nuclei of tumor cells [12,13]. Interestingly, we found that levels of phosphorylated FADD correlated with the proliferation capacity of tumor cells, supporting previous evidence in different cancer types [5,102].

As reviewed in [86], FADD has also emerged as a component of various signalosomes apart from the DISC and the necrosome, such as the FADDosome, the innateosome and the inflammasome. These can be initiated by death receptors, Toll-like receptors (TLR) or pathogen-induced signaling, and they can lead to pro-inflammatory signaling instead of cell death. The participation of FADD in signaling complexes eventually leading to gene expression changes has long been reported, particularly in relationship with innate immunity in mammals [128], *Drosophila* [129], or fish [130]. At the molecular level, this pathway seems to involve NF-κB activation. This pleiotropic transcription factor is present in almost all cell types and regulates the expression—either as an activator or as a repressor—of genes involved in many biological processes such as inflammation, immunity, differentiation, cell growth, and apoptosis. In consequence, aberrant or constitutive NF-κB activation impacts on various hallmarks of cancer as proliferation, migration or apoptosis and it has been detected in many tumor types both of hematological and solid nature [131]. Such hyperactivation of NF-κB signaling in cancer may be achieved by chromosomal alterations directly affecting members of the NF-κB pathway, as amplification of *c-Rel* on chromosome 2p14–15, chromosomal rearrangements or deletions affecting the *NF-κB2* locus on chromosome 10q24, t(14;19)(q32;q13) translocation resulting in increased expression of the transcriptional coactivator of p50 or p52 homodimers *Bcl-3*, and loss-of-function mutations in *IkBα* (reviewed in [131]). However, since *RelA*, *RelB* and *NF-kB1* alterations are rare in human cancer, different mechanisms might provide the source for NF-κB hyperactivation. This is the case with the tumor suppressor *PTEN* and *RAS* proto-oncogene, which are frequently mutated in cancer and whose abnormalities may in turn aberrantly activate NF-κB. However, it is also the case with NF-κB activation induced by death receptors or Toll-like receptors signaling. The formation of a “FADDosome” complex upon TRAIL-R signaling that promotes NF-κB activation and pro-inflammatory cytokine/chemokine production [132], operating both in nontransformed and transformed cells, has been demonstrated. In such a complex, FADD is the necessary adapter between caspase-8 and RIPK1. The transduction is critically dependent on caspase-8 in a protease-independent manner as a scaffold. Ubiquitin-modified RIPK1 would recruit proteins directly involved in activating NF-κB signaling, leading to the expression of inflammatory genes and eventually to the production of cytokines and chemokines. A similar “FADDosome” complex is, according to the literature, likely to occur upon CD95 signaling [133] and TLR signaling [134,135] in certain conditions. A frequent scenario in certain tumor types is the occurrence of *CASP-8* mutations that abolish the protease activity of the protein, rendering it unable to promote apoptosis. However, as they can bind to FADD, they can promote NF-κB activation upon appropriate death receptor or TLR stimuli [136,137]. This may act synergistically with the FADD overexpression observed in certain cancers. Related to this, increased levels of phosphorylated FADD have been linked to NF-κB hyperactivation in lung adenocarcinomas [5]. In this context, tumor cells would not only benefit from apoptosis resistance, but would also trigger tumorigenic pathways downstream NF-κB-mediated gene expression, such as inflammation or proliferation.

As discussed previously, the participation of FADD in the regulation of gene expression would be indirect, as a member of cytoplasmic protein complexes involved in transducing signals. Some authors have also proposed, however, that FADD may act in a more direct manner, as it is itself a member of the transcription factor complexes in the nucleus. In one study, it was speculated that FADD might be involved in regulating *MyoD* transcription [138]. The authors proposed that FADD would regulate the binding of SRF and/or MEF2 transcription factors to a *MyoD* enhancer, facilitating acetylation via recruitment of acetyltransferases such as CBP or p300, in turn activating gene transcription. Nevertheless, since FADD does not exhibit any DNA-binding motif, it might bind other proteins to form a functional transcription factor complex; in particular, the NF-κB activator NKAP has been proposed as a candidate to associate with FADD and regulate gene transcription in thymocytes [139]. NKAP has been shown to associate with HDAC3 as a part of a DNA-binding complex and, with CIR, as a part of the NOTCH co-repressor complex [140]. FADD association with NKAP in the nucleus would stabilize the latter, therefore contributing to its role as a transcriptional repressor able to hamper NOTCH-mediated transcriptional activation. High levels of NKAP have been observed in most tumor types (https://www.proteinatlas.org/ENSG00000101882-NKAP/pathology), and it has been described as an oncogene in hepatocellular carcinoma [141], glioma [142], and breast cancer [143], even as an unfavorable prognostic indicator in the latter (https://www.proteinatlas.org/ENSG00000101882-NKAP/pathology/tissue/breast+cancer). FADD downregulation in certain cancer types—such as T-LBL—might result in the reduced stability of NKAP complexes in the nucleus, contributing to NOTCH hyperactivation frequently found in T-cell lymphoblastic neoplasms [144]. However, there is no evidence for this association yet.

Besides its major role in cell death and the principal nonapoptotic functions described so far, comparative proteomics approaches have recently provided evidence for FADD’s involvement in energy metabolism [145,146,147,148]. In particular, the comparison of *FADD*-expressing with *FADD*-deficient mouse embryonic fibroblasts showed an enriched cluster of changed proteins involved in lipid metabolism, fatty acid metabolism, glycolysis, the tricarboxylic acid cycle, and oxidative phosphorylation [146,147]. The same authors proposed a mechanism whereby *FADD*-deficient thymocytes would exhibit reduced glucose uptake due to the accumulation of PKC-α, which may result in transcriptional repression of the glucose transporter gene *Glut1* [145]. Finally, this group has demonstrated in vivo in a mouse model that FADD is relevant to glucose and fat metabolism [148]. If FADD involvement in these functions is confirmed in humans, it could be speculated that the alterations in *FADD* observed in tumor cells may also impact on energy metabolism. The link between cancer and altered metabolism is well recognized, as cancer cells often utilize aerobic glycolysis, the well-known Warburg effect, to meet the needs of rapidly dividing neoplastic cells [149]. How *FADD* alterations would contribute to either metabolic switch in cancer still needs to be addressed.

## 4. Clinical Implications of *FADD* Alterations in Cancer

Considering the various alterations that FADD has been reported to undergo in pathological processes like cancer, many authors support the notion that the FADD level is a relevant factor to consider for diagnosis, prognosis, or therapeutic strategies. According to the information collected by The Human Protein Atlas, the *FADD* level is an unfavorable prognostic marker in lung, head and neck, and cervical cancers, whereas it constitutes a favorable prognostic marker in thyroid cancer (Figure 8).

In the literature, an association of *FADD* increase with poor clinical outcome has been described in many solid tumors [4,5,6,7,8,9,10,18,109,150,151,152]. In contrast, a reduction of *FADD* has been reported in fewer tumor types, among them hematological malignancies [11,12,13,14,15], but also in pathologies different from cancer like dementia [153]. Altered levels of phosphorylated FADD have also been reported, either a reduction [12,13,154,155,156] or an increase [5,7,37,98,100,101,102], but in all cases associated with poor outcome. This apparent discrepancy can probably be explained by the fact that FADD can play, as explained before, various roles depending on the cell type and molecular context. For example, our group demonstrated that both murine [12] and human [13] T-LBL samples exhibiting overall reduction of FADD and impaired apoptosis could be stratified according to their levels of phosphorylated FADD. S191/194-P-FADD was identified as an unfavorable clinical factor in T-LBL, suggesting that FADD phosphorylation status may serve as a new biomarker with prognostic value. On the contrary, another group reported that phosphorylated FADD was not prognostic in laryngeal carcinoma treated with radiotherapy [157], confirming that the predictive value of phosphorylated FADD also depends on cellular type and context.

Regarding the improvement of therapy, some interesting pieces of evidence concerning FADD are also available. For example, it has been reported that overexpressing *FADD* in colorectal cancer [158] or malignant glioma [159,160] renders tumor cells more sensitive to apoptosis, thus improving the effect of chemotherapy. In tumor types with frequent *FADD* amplification, such as head and neck cancer, it has been suggested that combined treatment with SMAC mimetics (Birinapant) and radiation may be particularly useful as FADD is key in sensitization to cell death [16]. Also, some chemotherapeutic agents—such as Carboplatin in tongue carcinoma [161] or Nortriptyline in bladder cancer cells [162]—induce the expression of *FADD*, thus contributing to tumor cell sensitization to apoptosis. FADD induction was observed in ovarian cancer cells upon combined treatment with cisplatin and AT-101, a natural BH3-mimetic molecule [163]. Interestingly, such treatment inhibited both DNMT and HDAC enzyme activities, allowing us to speculate that histone deacetylation might underlie the observed induction of *FADD* transcription. In chronic lymphocytic leukemia, it has been observed that the inhibition of histone deacetylase with Romidepsin exerts an apoptosis sensitization effect molecularly mediated by enhanced FADD recruitment to the DISC, although the exact mechanism is not clear [164]. In KRAS-driven lung cancer, it has been indicated that the inhibition of FADD phosphorylation suppresses tumor development, suggesting that FADD kinase is a plausible therapeutic target [102]. A summary of the *FADD* alterations reported in different cancer types and therapeutic agents described to target tumor cells through mechanisms involving *FADD* is depicted in Figure 9 and Figure 10, respectively.

## 5. Conclusions

The role of FADD in tumorigenesis is not fully understood. Its expression appears to be altered in many cancer types, but the underlying mechanisms are not always clear. Chromosomal alterations affecting *FADD* are more frequent than mutations or polymorphisms, but they do not always explain the changes in expression observed in tumor cells. Regulation of *FADD* expression by transcription factors and epigenetic mechanisms may also be altered in cancer, but limited evidence is available to confirm this. Furthermore, posttranslational modifications of FADD, especially phosphorylation and ubiquitination, may underlie the alterations of FADD functions in tumor cells. Besides its canonical role as an adaptor for cell death, FADD has been implicated in various signalosomes that would trigger tumorigenic pathways downstream of NF-κB-mediated gene expression, such as inflammation or proliferation. A role for FADD in directly regulating gene expression as a member of protein complexes in the nucleus has also been proposed, although the available evidence is still scarce. In addition, the implication of FADD in controlling energy metabolism is mainly based on recent evidence of protein‒protein interactions. In any case, many authors agree that FADD levels are a relevant factor to consider in the clinical management of cancer. However, discrepant results suggest that its role as a positive or negative prognostic factor would depend on the tumor type and cellular context. Future research will likely shed light on the molecular details of FADD functions, thus clarifying its clinical value for each cancer type.

## Figures and Tables

**Figure 1 cancers-11-01462-f001:**
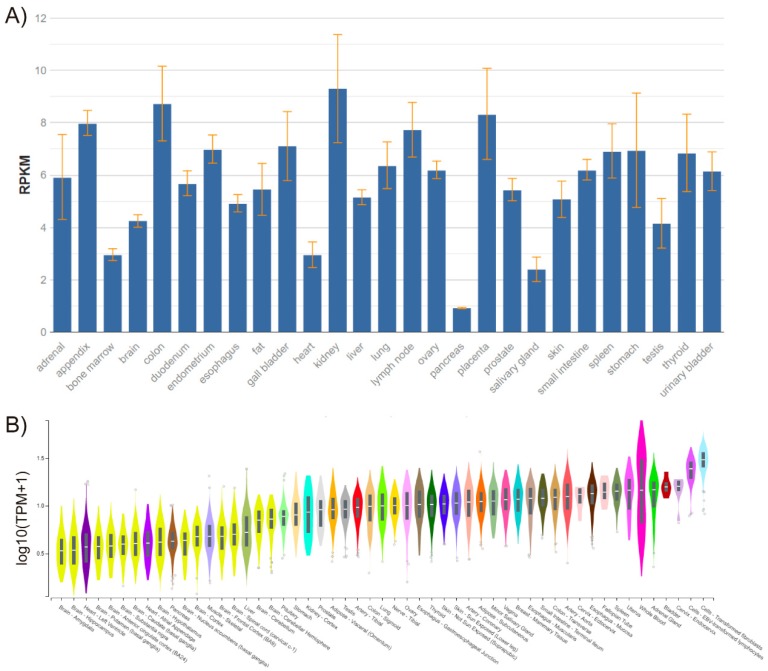
*FADD* expression in healthy human tissues, performed by RNA sequencing. (**A**) Data obtained from BioProject PRJEB4337 of samples from 95 individuals representing 27 different healthy tissues [3]. RPKM, Reads Per Kilobase Million. (**B**) Data obtained from GTEx Analysis Release V7 (dbGaP Accession phs000424.v7.p2). Expression values are shown in transcripts per million (TPM), calculated from a gene model with isoforms collapsed to a single gene and no other normalization steps applied. Box plots are shown as median and 25th and 75th percentiles; points are displayed as outliers if they are above or below 1.5 times the interquartile range.

**Figure 2 cancers-11-01462-f002:**
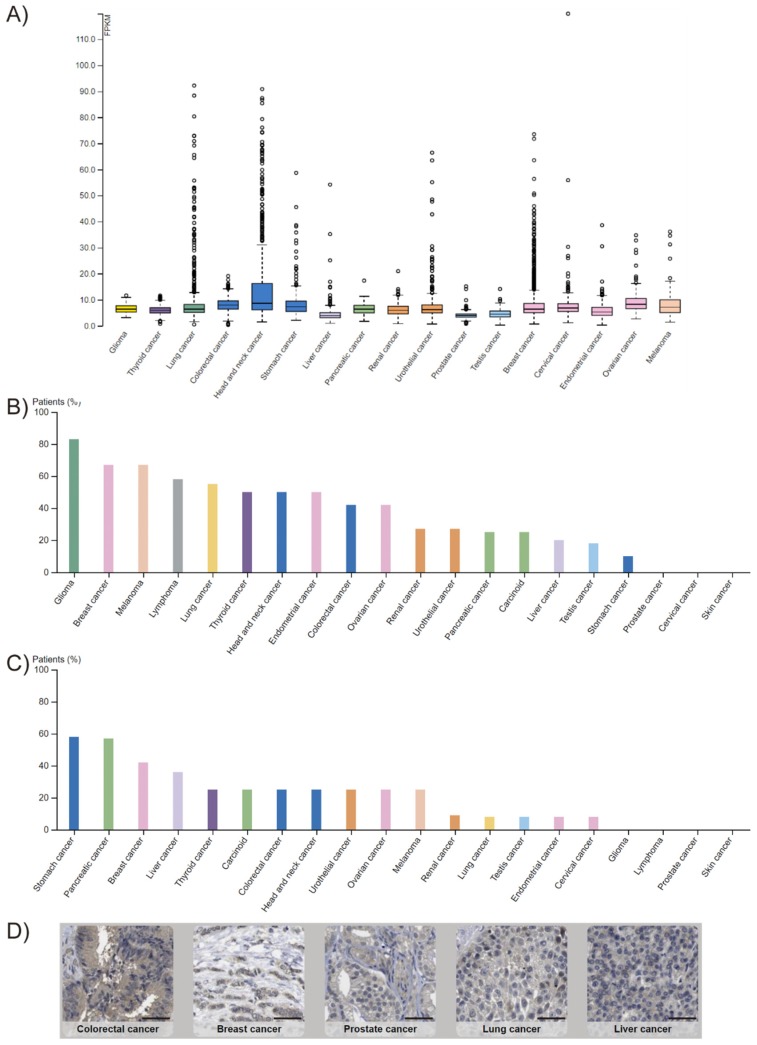
*FADD* expression in cancer. (**A**) RNA-sequencing data from The Cancer Genome Atlas (TCGA) project of Genomic Data Commons (GDC). Seventeen cancer types representing 21 cancer subtypes with a corresponding major cancer type in The Human Pathology Atlas were included to allow for comparisons with the protein staining data from The Human Protein Atlas. The FPKMs (number fragments per kilobase of exon per million reads) were used for quantification of expression with a detection threshold of 1 FPKM. (**B**,**C**) FADD protein levels from The Human Protein Atlas. For each cancer, color-coded bars indicate the percentage of patients (maximum: 12) with high and medium protein expression level. Low or not detected protein expression results in a white bar. (**B**) Results obtained using HPA001464 antibody. Cases of colorectal, breast, ovarian, urothelial, gastric, pancreatic, and liver cancers, and melanomas showed weak to moderate cytoplasmic positivity. The remaining cancers were negative. (**C**) Results obtained using CAB010209 antibody. The majority of cancer cells showed weak to moderate cytoplasmic positivity. Nucleolar staining was observed in several cases. A few breast cancers were strongly stained. (**D**) Selection of five standard cancer tissue samples representative of the overall staining pattern. Scale bar: 50 µm.

**Figure 3 cancers-11-01462-f003:**
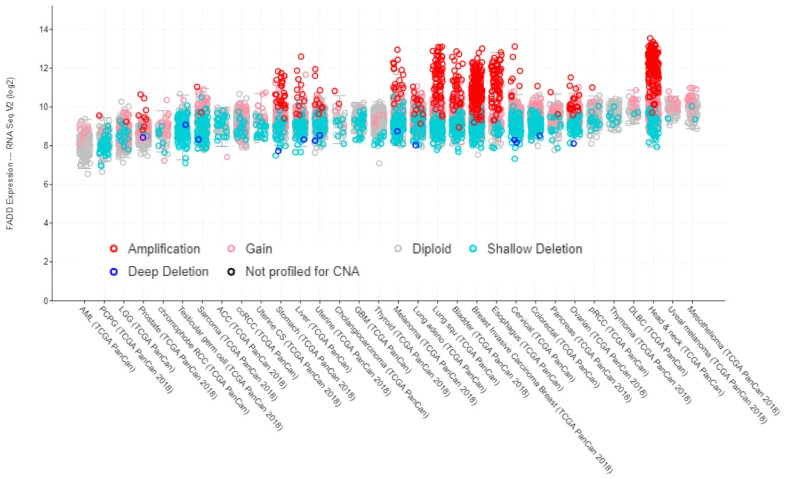
*FADD* expression in human cancer, indicating the occurrence of copy number alterations. Screenshot image modified from cBioPortal. RNA-sequencing data are obtained from TCGA PanCancer Atlas Studies in 10,967 tumor samples from various origins.

**Figure 4 cancers-11-01462-f004:**
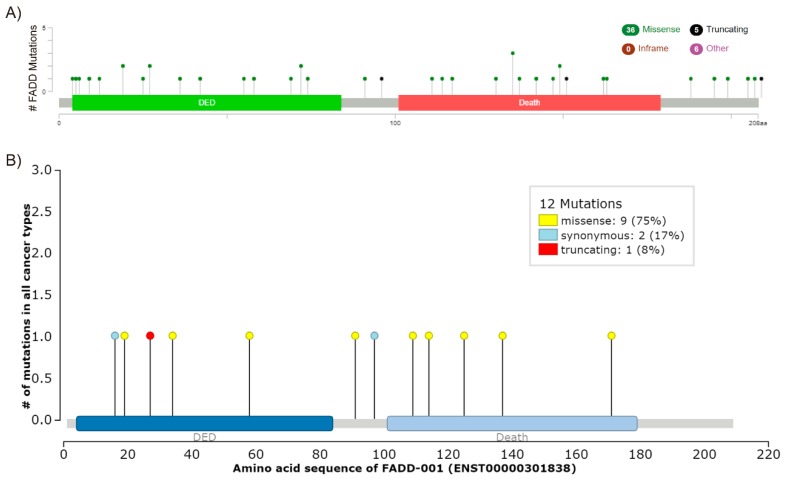
*FADD* mutations. (**A**) Screenshot modified from cBioPortal. Mutation diagram circles are colored with respect to the corresponding mutation types. In case of different mutation types at a single position, the color of the circle is determined with respect to the most frequent mutation type. (**B**) Screenshot image modified from IntOGen. The mutations needle plot shows the distribution of the observed cancer mutations along the protein sequence and its possible mutational clusters and hotspots. The needles’ height and head size represent mutational recurrence. Needles of different categories that fall in the same amino acid residues are stacked.

**Figure 5 cancers-11-01462-f005:**
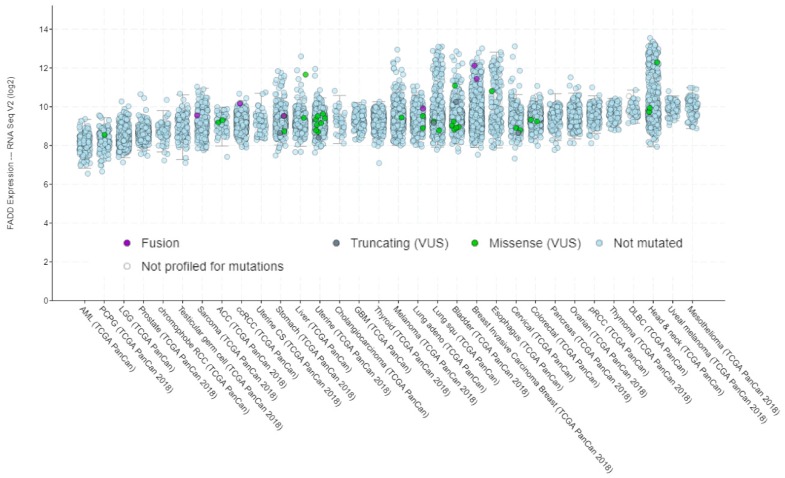
*FADD* expression in human cancer, indicating the occurrence of mutations. Screenshot image modified from cBioPortal. RNA-sequencing data are obtained from TCGA PanCancer Atlas Studies in 10967 tumor samples from various origins.

**Figure 6 cancers-11-01462-f006:**
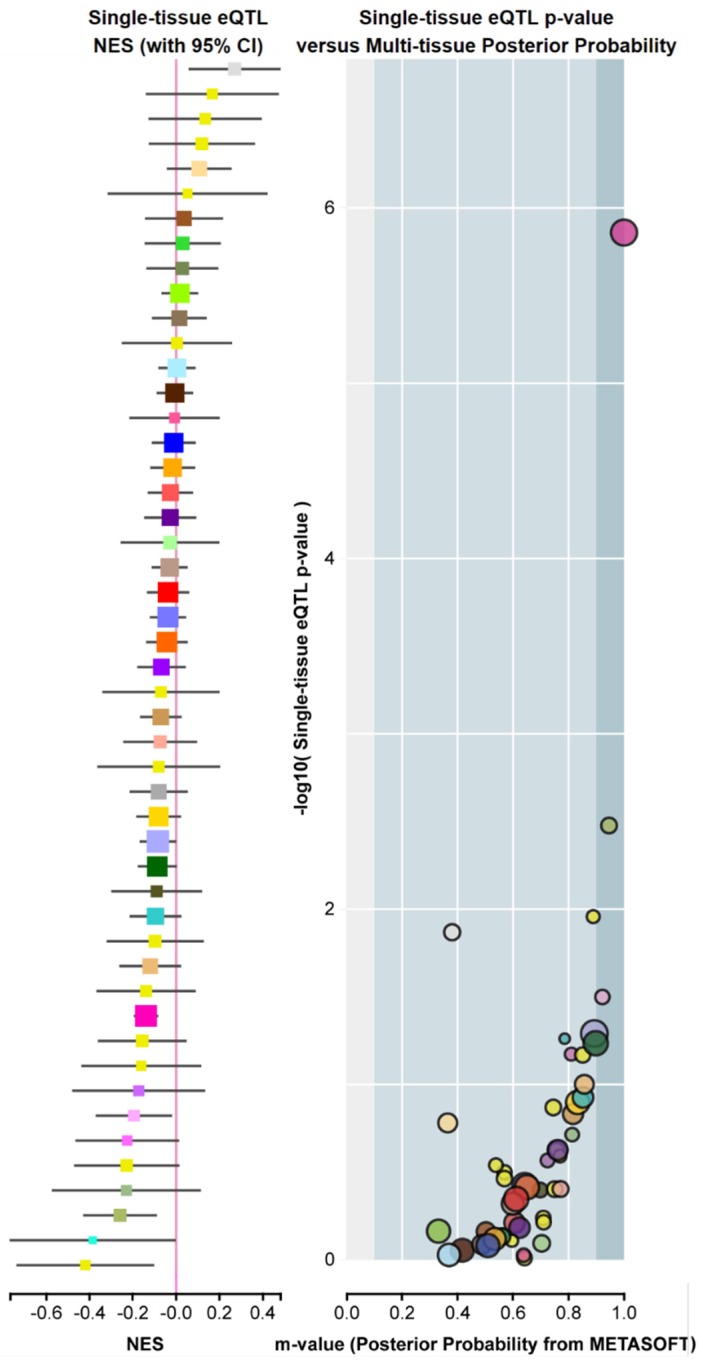
Example of *FADD* expression quantitative trait loci (eQTLs), or *FADD* genetic variants exhibiting high correlation with changes in gene expression. Screenshot from GTEx Portal showing, for variant ID chr11_70241122_G_GCTT_b38, SNP rs559543475, single-tissue eQTL normalized effect size (NES) with 95% confidence interval (left) and Single-tissue eQTL *p*-value vs. Multi-tissue posterior probability (right). The *Y* axis indicates the −log_10_ of *p*-value (obtained from a *t*-test that compares observed beta from single-tissue eQTL analysis to a null beta of 0). The *X* axis indicates the *m*-value, which indicates the posterior probability that an eQTL effect exists in each tissue tested in the cross-tissue meta-analysis. The m-value ranges between 0 and 1 and is interpreted as follows: *m*-value < 0.1 indicates that the tissue is predicted to not have an eQTL effect; *m*-value > 0.9 indicates that the tissue is predicted to have an eQTL effect; otherwise, the prediction of the existence of an eQTL effect is ambiguous [58]. Normalized effect size (NES): the slope of the linear regression of normalized expression data versus the three genotype categories using single-tissue eQTL analysis, representing eQTL effect size. The normalized expression values are based on quantile normalization within each tissue, followed by inverse quantile normalization for each gene across samples. Colors represent distinct tissue categories.

**Figure 7 cancers-11-01462-f007:**
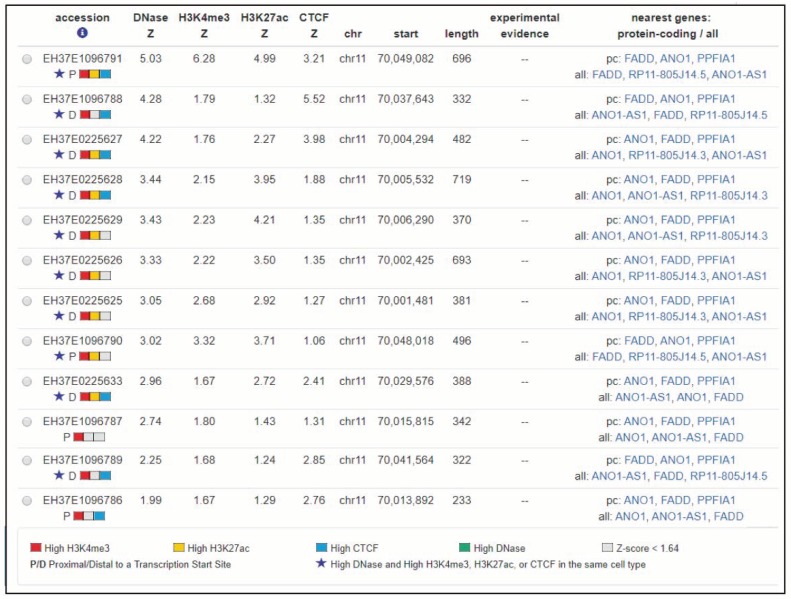
Screenshot image from SCREEN hg19 (search candidate cis-regulatory elements by ENCODE). This search is showing candidate cis-regulatory elements located between the first and last transcription start sites (TSSs) of *FADD* and up to 50 kb upstream.

**Figure 8 cancers-11-01462-f008:**
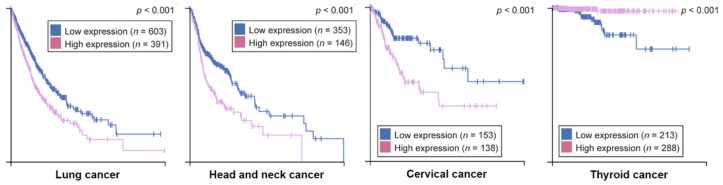
Kaplan‒Meier plots for cancer types where high *FADD* expression has significant (*p* < 0.001) association with patient survival. Based on the FPKM value, patients were divided based on *FADD* mRNA level into one of the two groups “low” (under cutoff) or “high” (over cutoff). The *X*-axis shows time for survival (years) and the *Y*-axis shows the probability of survival, where 1.0 corresponds to 100 percent. The prognosis of each group of patients was examined by Kaplan‒Meier survival estimators, and the survival outcomes of the two groups were compared by log-rank tests. (From The Human Protein Atlas).

**Figure 9 cancers-11-01462-f009:**
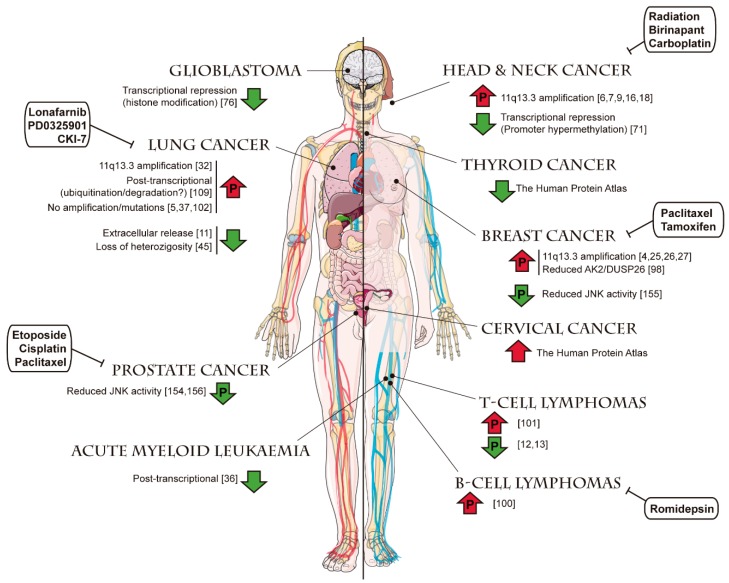
Alterations of *FADD* levels reported in different cancer types. For each tumor type, reported evidence of an increase (upwards red arrow) or a decrease (downwards green arrow) of *FADD* levels is shown, indicating the underlying mechanism when this information is available (references indicated in brackets). Arrows including a “P” indicate changes also affecting the levels of phosphorylated FADD. Anticancer therapies reported to target *FADD* in certain tumor types are shown in white boxes, and their mechanisms of action are detailed in Figure 10. Templates to build this figure were obtained from SMART Servier Medical Art (Attribution 3.0 Unported, CC BY 3.0).

**Figure 10 cancers-11-01462-f010:**
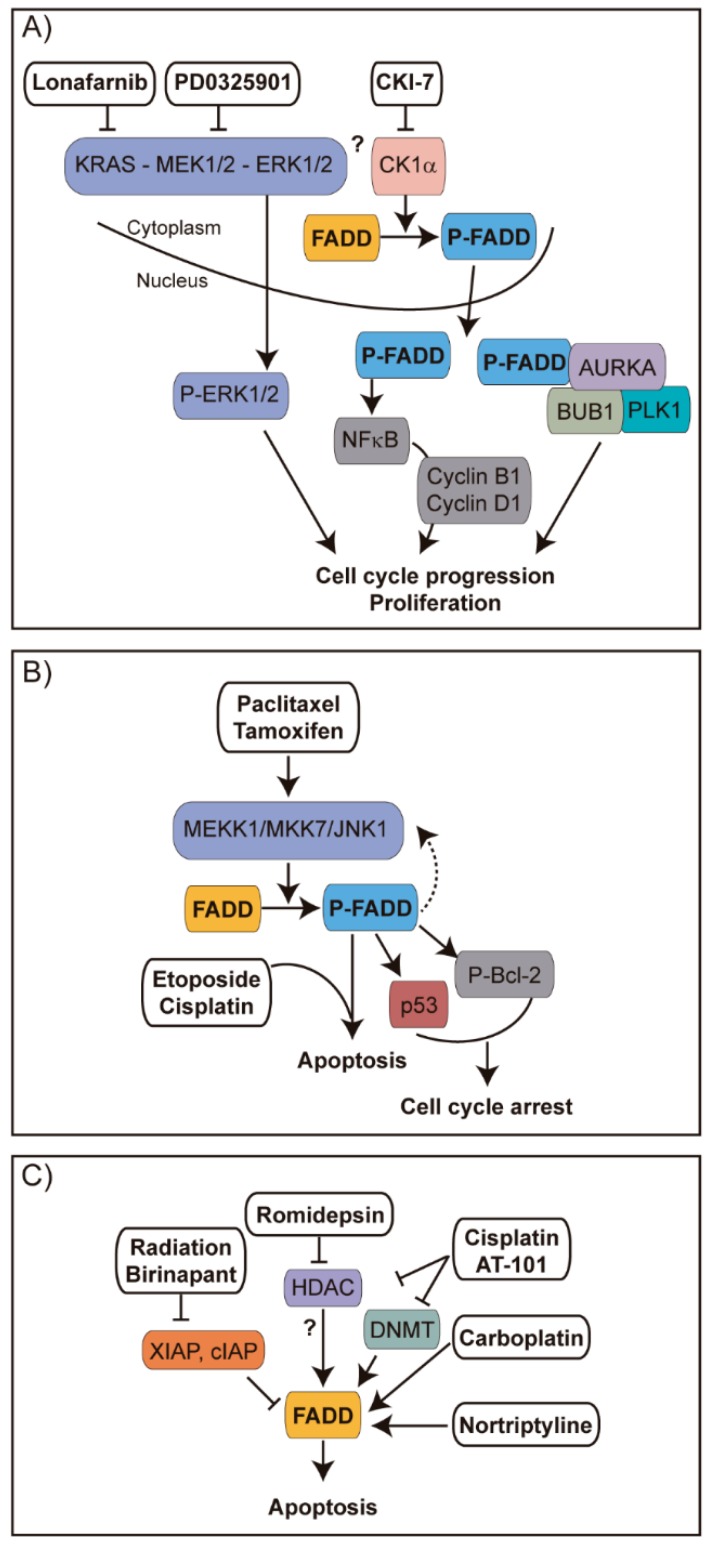
Strategies for therapeutic intervention of cancer involving FADD. (**A**) In lung cancer, CKIα-mediated phosphorylation of FADD leads to the translocation of S194-P-FADD to the nucleus, where it can induce expression of cyclins B1 and D1 through NF-κB signaling [5], or it can interact with key G2/M transition proteins like AURKA, PLK1 or BUB1 [102], promoting cell cycle progression and proliferation. Inhibiting KRAS with Lonafarnib, MEK with PD0325901, or CK1α with CKI-7 decreased the abundance of phosphorylated FADD and decreased cell proliferation, apparently due to a loss of interaction between FADD and the G2/M transition proteins. As a result, tumor treated cells would fail to progress through G2/M. (**B**) Tamoxifen and paclitaxel in breast cancer [155] and paclitaxel in prostate cancer [165] have been reported to activate the MEKK1/MKK7/JNK pathway, which contributes to FADD phosphorylation. In breast cancer, this results in cell cycle arrest and suppression of cancer growth through p53 stabilization or Bcl-2 phosphorylation. In prostate cancer, phosphorylated FADD in turn upregulates MEKK1 and downstream JNK1 activation, which is essential for sensitization to apoptosis induced by etoposide or cisplatin combined with paclitaxel [165]. Thus, chemosensitization can be amplified through FADD phosphorylation and the MEKK1/MKK7/JNK1 pathway. (**C**) In head and neck cancer, the SMAC mimetic Birinapant plus radiation induces tumor regression [16]. Radiation induces DNA damage and in consequence the intrinsic cell death pathway through mitochondrial release of SMAC. The negative effect of SMAC on IAPs is enhanced by SMAC mimetic Birinapant, resulting in degradation of IAPs to enhance FADD-involving DR-induced apoptosis. In chronic lymphocytic leukemia, the HDAC inhibitor Romidepsin sensitizes tumor cells to TRAIL-induced apoptosis through enhancement of FADD recruitment to the DISC [164]. In ovarian cancer, the combination of the BH3-mimetic molecule AT-101 with cisplatin strongly sensitize cells towards apoptosis; they inhibit HDAC and DNA methyltransferase (DNMT) enzyme activities and they induce *FADD* expression among other apoptosis-related genes [163]. Carboplatin and Nortriptyline also favor *FADD* expression in tongue carcinoma [161] and bladder cancer cells [162], respectively.

**Table 1 cancers-11-01462-t001:** ClinVar search results for *FADD* [gene] and “single gene” [properties].

Name	Condition(s)	Clinical Significance (Last Reviewed)	GRCh37 Location	GRCh38 Location	Variation ID	Allele ID(s)
NM_003824.3(FADD): c.31G > A (p.Val11Met)	Infections, recurrent, with encephalopathy, hepatic dysfunction, and cardiovascular malformations	Uncertain significance (26 February 2018)	11: 70049596	11: 70203490	579410	566162
NM_003824.3(FADD): c.93G > T (p.Val31=)		Benign (30 October 2017)	11: 70049658	11: 70203552	471687	461658
NM_003824.3(FADD): c.168G > T (p.Glu56Asp)		Uncertain significance (22 September 2017)	11: 70049733	11: 70203627	539062	526519
NM_003824.3(FADD): c.287-8C > G		Likely benign (21 July 2017)	11: 70052231	11: 70206125	471685	462293
NM_003824.3(FADD): c.315T > G (p.Cys105Trp)		Pathogenic (10 December 2010)	11: 70052267	11: 70206161	30267	39223
NM_003824.3(FADD): c.452C > T (p.Thr151Ile)		Uncertain significance (26 June 2018)	11: 70052404	11: 70206298	575179	564906
NM_003824.3(FADD): c.475G > A (p.Ala159Thr)		Uncertain significance (3 July 2017)	11: 70052427	11: 70206321	471686	461663

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
