# Peer review of "FADD in Cancer: Mechanisms of Altered Expression and Function, and Clinical Implications"

_cancers, 2019, doi:10.3390/cancers11101462_

Round 1

Reviewer 1 Report

The manuscript is a review bringing together information on the FADD gene and its encoded protein. The authors have described genetic changes affecting the gene, post-translational changes affecting the protein product as well as potential oncogenic mechanisms.  

The review is well organized and is extremely useful in bringing the extensive information on FADD into a concise article for the researcher.

Major comment

Given the title of the review, the manuscript would be improved by the expansion of Section 5 on the clinical implications of FADD in cancer. For example, the provision of a diagram showing possible affected pathways in different tumour types such as haematopoetic or solid tumours. This information could then be linked to the effects given by various therapeutic agents.   

Minor comments

The use of English should be checked throughout the manuscript. There  are many instances where the sentences lack clarity. Examples include:

Page 8 - Line 215 - '.....conform the histone code, which can be implicated...."

Page 11 - Line 328 "....FADD ability..."

Page 11 - Line 345 - "....been reported since long....."

Page 13 - line 395 -".... FADD has been reported in less tumour types" where 'less' should be replaced by 'fewer.'

Page 13 - Line 397 "...resulting reduced (refs) or increased (refs......." does not make sense.

Page 13 - Line 423 - The sentence "The complexity of FADD implicaion in cancer is evident" does not make sense. 

Care should be taken throughout the manuscript to use FADD and Fadd for human and mouse proteins respectively and FADD and Fadd for the human and mouse genes, respectively. The terms appear to be used interchangeably. Examples include Page 8 line 203 - "Methylation of FADD  promoter region" Should this be Methylation of FADD promoter region? Also, the following paragraph is unclear as to whether it is the gene or its product that are being described. 'fadd' is also used in section headings such as on Page 10- lines 258 and 299, Page 1 -lines 337 and 343. 

Reviewer 2 Report

This review covers the potential roles of FADD in cancer. The review covers a lot of ground but in a number of sections I feel may be a bit too speculative, or does not relate the section back to a role of FADD in Cancer and as such comes of more as a literature review without adding context to the literature cited. This could be addressed by relating the cited information more precisely back to the potential roles in Cancer, as the authors are proposing FADD probably has.

For example, the Role of fadd in regulation of gene expression section is very short, of course not a lot is known as the authors suggest, but the section could be improved by relating these ideas to questions such as is NF-kB important in driving cancer? what kinds of cancer etc. some follow up as opposed to just listing the possible roles of fadd in gene regulation would be more insightful and greatly improve the review I think. In addition this section could also include activation of NF-kB in response to death receptors such as TRIAL etc. as it has been clearly shown (Henry, C. M. & Martin, S. J. Caspase-8 Acts in a Non-enzymatic Role as a Scaffold for Assembly of a Pro-inflammatory ‘FADDosome’ Complex upon TRAIL Stimulation. Mol Cell 65, 715–729.e5 (2017).) to require caspase-8 scaffolding, which may also be important.

Similar comments could be made about a number of the sections and providing a more directed commentary about how FADD overexpression could help drive tumour growth etc. would make the review a more useful resource.

Have the authors considered including data form the Human Protein Atlas (https://www.proteinatlas.org/ENSG00000168040-FADD/pathology) to show expression at the protein level in different cancers. This may add another level of association if they would like to make the point the FADD could have prognostic value.

More specific comments:

While I appreciate that there are many reviews covering the function of FADD in death receptor signalling, I do not think this means that this should be left completely uncovered in the review. The topic warrants, at the very least, a short summary of the function of FADD as an adapter for receptor signalling. This would help the reader, especially those not well read in the field of death receptor and TLR signalling.

Page 2 Line 57-60: the authors state “Furthermore, this group and others have demonstrated the association between FADD amplification and high FADD levels with poor overall survival and disease free survival, especially if both events occur together [6,17], supporting the role of FADD as a driver of 11q13.3 amplification.”

I believe the authors are trying to say the it supports FADD as the driver of the tumourigenic effects of the 11q13.3 amplification, as I do not see how these studies can claim that FADD is itself driving the duplication.

Page 8 Line 197-198: the authors state “Altogether, this functional link points at a prominent role of FADD alteration in coloboma.”

While this association may hint that FADD plays a role in colomboma, to say that it “points to a prominent role” is I think a bit strong, given it is still speculative. I suggest the wording should be changed to be less strong.

Figure 6: A small point, but could this figure be made a bit nicer by removing the shopping cart symbols in it?

Page 11: line 310-311: the authors state “FADD binds directly to RIP1, thus inhibiting the interaction RIP1-RIP3 and blocking necroptosis in consequence [114-117]”.

This is incorrect. FADD allows the caspase-8/cFLIP heterodimer to be recruited to RIPK1/3 thus cleaving their kinase domain away and inactivating them, thus blocking necroptosis. It is not blocking RIPK3 recruitment which binds RIPK1 through the RHIM motif. Please correct this section to reflect this.

Page 11: lines 322-336: This paragraph discusses the role of FADD in T-cell proliferation, making the point the FADD is required for cell cycle progression, as well as survival. I believe it is well accepted now that loss of FADD or caspase-8 etc. in T-cells leads to RIPK1 dependent cell death, either apoptosis or necroptotis, Dead cells can obviously not go through the cell cycle, so I think this section is a bit misleading and should be adjusted to make this clear.

The language could use some careful checking as well.

Round 2

Reviewer 2 Report

The authors have addressed the previously raised concerns and the manuscript is greatly improved.  It will be a useful resource.

A minor issue I have noticed that the placement of the references seems to have gone askew in a few places (example page 14 line 455). The heading starts with references. this occurs in a few places such as sentences starting with references and so on. This should be checked carefully.

Also the new Figure 9. If the authors have made this figure completely themselves then congratulations. It is a very nice drawing. If it has been adapted from another source, then please reference it and make sure you have permission to use it.
